# SCALING PARETO-EFFICIENT DECISION MAKING VIA OFFLINE MULTI-OBJECTIVE RL

**Baiting Zhu, Meihua Dang, Aditya Grover**
University of California, Los Angeles, CA, USA
`baitingzbt@g.ucla.edu, mhdang@cs.ucla.edu, adityag@cs.ucla.edu`

## ABSTRACT

The goal of multi-objective reinforcement learning (MORL) is to learn policies that simultaneously optimize multiple competing objectives. In practice, an agent's preferences over the objectives may not be known apriori, and hence, we require policies that can generalize to arbitrary preferences at test time. In this work, we propose a new data-driven setup for *offline* MORL, where we wish to learn a preference-agnostic policy agent using only a finite dataset of offline demonstrations of other agents and their preferences. The key contributions of this work are two-fold. First, we introduce D4MORL, (D)datasets for MORL that are specifically designed for offline settings. It contains 1.8 million annotated demonstrations obtained by rolling out reference policies that optimize for randomly sampled preferences on 6 MuJoCo environments with 2-3 objectives each. Second, we propose Pareto-Efficient Decision Agents (PEDA), a family of offline MORL algorithms that builds and extends return-conditioned offline methods including Decision Transformers (Chen et al., 2021) and RvS (Emmons et al., 2021) via a novel preference-and-return conditioned policy. Empirically, we show that PEDA closely approximates the behavioral policy on the D4MORL benchmark and provides an excellent approximation of the Pareto-front with appropriate conditioning, as measured by the hypervolume and sparsity metrics.

## 1 INTRODUCTION

We are interested in learning agents for multi-objective reinforcement learning (MORL) that optimize for one or more competing objectives. This setting is commonly observed in many real-world scenarios. For instance, an autonomous driving car might trade off high speed and energy savings depending on the user's preferences. If the user has a relatively high preference for speed, the agent will move fast regardless of power usage; on the other hand, if the user tries to save energy, the agent will keep a more steady speed. One key challenge with MORL is that different users might have different preferences on the objectives and systematically exploring policies for each preference might be expensive, or even impossible. In the online setting, prior work considers several approximations based on scalarizing the vector-valued rewards of different objectives based on a single preference (Lin, 2005), learning an ensemble of policies based on enumerating preferences (Mossalam et al., 2016, Xu et al., 2020), or extensions of single-objective algorithms such as Q-learning to vectorized value functions (Yang et al., 2019).

We introduce the setting of *offline* multi-objective reinforcement learning for high-dimensional state and action spaces, where our goal is to train an MORL policy agent using an offline dataset of demonstrations from multiple agents with known preferences. Similar to the single-task setting, offline MORL can utilize auxiliary logged datasets to minimize interactions, thus improving data efficiency and minimizing interactions when deploying agents in high-risk settings. In addition to its practical utility, offline RL (Levine et al., 2020) has enjoyed major successes in the last few years (Kumar et al., 2020, Kostrikov et al., 2021, Chen et al., 2021) on challenging high-dimensional environments for continuous control and game-playing. Our contributions in this work are two-fold in introducing benchmarking datasets and a new family of MORL, as described below.

We introduce Datasets for Multi-Objective Reinforcement Learning (D4MORL), a collection of 1.8 million trajectories on 6 multi-objective MuJoCo environments (Xu et al., 2020). Here, 5 environ-

ments consist of 2 objectives and 1 environment consists of 3 objectives. For each environment in D4MORL, we collect demonstrations from 2 pretrained behavioral agents: *expert* and *amateur*, where the relative expertise is defined in terms of the Pareto-efficiency of the agents and measured empirically via their hypervolumes. Furthermore, we also include 3 kinds of preference distributions with varying entropies to expose additional data-centric aspects for downstream benchmarking. Lack of MORL datasets and large-scale benchmarking has been a major challenge for basic research (Hayes et al., 2022), and we hope that D4MORL can aid future research in the field.

Next, we propose Pareto-Efficient Decision Agents (PEDA), a family of offline MORL algorithms that extends return-conditioned methods including Decision Transformer (DT) (Chen et al., 2021) and RvS (Emmons et al., 2021) to the multi-objective setting. These methods learn a return-conditioned policy via a supervised loss on the predicted actions. In recent work, these methods have successfully scaled to agents that demonstrate broad capabilities in multi-task settings (Lee et al., 2022 Reed et al., 2022). For MORL, we introduce a novel preference and return conditioned policy network and train it via a supervised learning loss. At test time, naively conditioning on the default preferences and maximum possible returns leads to out-of-distribution behavior for the model, as neither has it seen maximum returns for all objectives in the training data nor is it possible to simultaneously maximize all objectives under competition. We address this issue by learning to map preferences to appropriate returns and hence, enabling predictable generalization at test-time.

Empirically, we find PEDA performs exceedingly well on D4MORL and closely approximates the reference Pareto-frontier of the behavioral policy used for data generation. In the multi-objective HalfCheetah environment, compared with an average upper bound on the hypervolume of $5.79 \times 10^6$ achieved by the behavioral policy, PEDA achieves an average hypervolume of $5.77 \times 10^6$ on the `Expert` and $5.76 \times 10^6$ on the `Amateur` datasets.

## 2 RELATED WORK

**Multi-Objective Reinforcement Learning** Predominant works in MORL focus on the online setting where the goal is to train agents that can generalize to arbitrary preferences. This can be achieved by training a single preference-conditioned policy (Yang et al., 2019; Parisi et al., 2016), or an ensemble of single-objective policies for a finite set of preferences (Mossalam et al., 2016; Xu et al., 2020; Zhang & Li, 2007). Many of these algorithms consider vectorized variants of standard algorithms such as Q-learning (Mossalam et al., 2016; Yang et al., 2019), often augmented with strategies to guide the policy ensemble towards the Pareto front using evolutionary or incrementally updated algorithms (Xu et al., 2020; Zhang & Li, 2007; Mossalam et al., 2016; Roijers et al., 2014; Huang et al., 2022). Other approaches have also been studied, such as framing MORL as a meta-learning problem (Chen et al., 2019), learning the action distribution for each objective (Abdolmaleki et al., 2020), and learning the relationship between objectives (Zhan & Cao, 2019) among others. In contrast to these online MORL works, our focus is on learning a single policy that works for all preferences using only offline datasets.

There are also a few works that study decision-making with multiple objectives in the offline setting and sidestep any interaction with the environments. Wu et al., 2021 propose a provably efficient offline MORL algorithm for tabular MDPs based on dual gradient ascent. Thomas et al., 2021 study learning of safe policies by extending the approach of Laroche et al., 2019 to the offline MORL setting. Their proposed algorithm assumes knowledge of the behavioral policy used to collect the offline data and is demonstrated primarily on tabular MDPs with finite state and action spaces. In contrast, we are interested in developing dataset benchmarks and algorithms for scalable offline policy optimization in high-dimensional MDPs with continuous states and actions.

**Multi-Task Reinforcement Learning** MORL is also closely related to multi-task reinforcement learning, where every task can be interpreted as a distinct objective. There is an extensive body of work in learning multi-task policies both in the online and offline setups (Wilson et al., 2007; Lazaric & Ghavamzadeh, 2010; Teh et al., 2017) inter alia. However, the key difference is that typical MTRL benchmarks and algorithms do not consider solving multiple tasks that involve inherent trade-offs. Consequently, there is no notion of Pareto efficiency and an agent can simultaneously excel in all the tasks without accounting for user preferences.

**Reinforcement Learning Via Supervised Learning**  A body of recent works have formulated off-line reinforcement learning as an autoregressive sequence modeling problem using Decision Transformers (DT) or Trajectory Transformers ( Chen et al., 2021, Janner et al., 2021) The key idea in DT is to learn a transformer-based policy that conditions on the past history and a dynamic estimate of the returns (a.k.a. returns-to-go). Follow-up works consider online learning (Zheng et al., 2022) as well as simpler variants that rely only on multi-layer perceptrons (Emmons et al., 2021). Such agents are generally more stable and robust to optimize due to the simplicity of loss function and easier to scale to more complex settings such as environments with high-dimensional actions or states, as shown in recent works in multi-task RL (Lee et al., 2022; Reed et al., 2022).

## 3 PRELIMINARIES

**Setup and Notation.**  We operate in the general framework of a multi-objective Markov decision process (MOMDP) with linear preferences (Wakuta, 1995). An MOMDP is represented by the tuple $\langle \mathcal{S}, \mathcal{A}, \mathcal{P}, \mathcal{R}, \Omega, f, \gamma \rangle$. At each timestep $t$, the agent with a current state $s_t \in \mathcal{S}$ takes an action $a_t \in \mathcal{A}$ to transition into a new state $s_{t+1}$ with probability $\mathcal{P}(s_{t+1}|s_t, a_t)$ and observes a reward vector $r_t = \mathcal{R}(s_t, a_t) \in \mathbb{R}^n$. Here, $n$ is the number of objectives. The vector-valued return $\mathbf{R} \in \mathbb{R}^n$ of an agent is given by the discounted sum of reward vectors over a time horizon, $\mathbf{R} = \sum_t \gamma^t r_t$. We also assume that there exists a linear utility function $f$ and a space of preferences $\Omega$ that can map the reward vector $r_t$ and a preference vector $\omega \in \Omega$ to a scalar reward $r_t$, i.e., $r_t = f(r_t, \omega) = \omega^\intercal r_t$. The expected vector return of a policy $\pi$ is given an $\mathbf{G}^\pi = [G_1^\pi, G_2^\pi, \ldots, G_n^\pi]^\intercal$ where the expected return of the $i^{\text{th}}$ objective is given as $G_i^\pi = \mathbb{E}_{a_{t+1} \sim \pi(\cdot|s_t, \omega)}[\sum_t \mathcal{R}(s_t, a_t)_i]$ for some predefined time horizon and preference vector $\omega$. The goal is to train a multi-objective policy $\pi(a|s, \omega)$ such that the expected scalarized return $\omega^\intercal \mathbf{G}^\pi = \mathbb{E}[\omega^\intercal \sum_t \mathcal{R}(s_t, a_t)]$ is maximized.

**Pareto Optimality.**  In MORL, one cannot optimize all objectives simultaneously, so policies are evaluated based on the *Pareto set* of their vector-valued expected returns. Consider a preference-conditioned policy $\pi(a|s, \omega)$ that is evaluated for $m$ distinct preferences $\omega_1, \ldots, \omega_m$, and let the resulting policy set be represented as $\{\pi_p\}_{p=1,\ldots,m}$, where $\pi_p = \pi(a|s, \omega = \omega_p)$, and $\mathbf{G}^{\pi_p}$ is the corresponding unweighted expected return. We say the solution $\mathbf{G}^{\pi_p}$ is *dominated* by $\mathbf{G}^{\pi_q}$ when there is no objective for which $\pi_q$ is worse than $\pi_p$, i.e., $G_i^{\pi_p} < G_i^{\pi_q}$ for $\forall i \in [1, 2, \ldots, n]$. If a solution is not dominated, it is part of the Pareto set denoted as $P$. The curve traced by the solutions in a Pareto set is also known as the Pareto front. In MORL, our goal is to define a policy such that its empirical Pareto set is a good approximation of the true Pareto front. While we do not know the true Pareto front for many problems, we can define metrics for relative comparisons between different algorithms. Specifically, we evaluate a Pareto set $P$ based on two metrics, *hypervolume* and *sparsity* that we describe next.

**Definition 1** (Hypervolume). Hypervolume $\mathcal{H}(P)$ measures the space or volume enclosed by the solutions in the Pareto set $P$:

$$\mathcal{H}(P) = \int_{\mathbb{R}^m} \mathbb{1}_{H(P)}(z)\, dz,$$

where $H(P) = \{z \in Z | \exists i : 1 \leq i \leq |P|, r \leq z \leq P(i)\}$. $P(i)$ is the $i^{\text{th}}$ solution in $P$, $\preceq$ is the dominance relation operator, and $\mathbb{1}_{H(P)}(z)$ equals 1 if $z \in H(P)$ and 0 otherwise. Higher hypervolumes are better.

**Definition 2** (Sparsity). Sparsity $\mathcal{S}(P)$ measures the density of the Pareto front covered by a Pareto set $P$:

$$\mathcal{S}(P) = \frac{1}{|P|-1} \sum_{i=1}^{n} \sum_{k=1}^{|P|-1} (\tilde{P}_i(k) - \tilde{P}_i(k+1))^2,$$

where $\tilde{P}_i$ represents a list sorted as per the values of the $i^{\text{th}}$ objective in $P$ and $\tilde{P}_i(k)$ is the $k^{\text{th}}$ value in the sorted list. Lower sparsity is better.

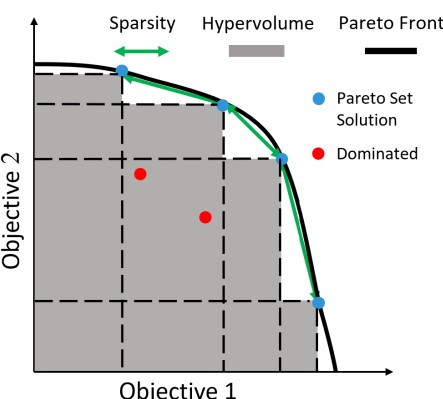

Figure 1: Illustration of the Hypervolume and Sparsity Metrics. Only undominated solutions (i.e., the Pareto set) are used for calculating the evaluation metrics.

See Figure 1 for an illustration and Appendix F for discussion on other possible metrics.

# 4  D4MORL: DATASETS FOR OFFLINE MULTI-OBJECTIVE REINFORCEMENT LEARNING

In offline RL, the goal of an RL agent is to learn the optimal policy using a fixed dataset without any interactions with the environment (Levine et al., 2020). This perspective brings RL closer to supervised learning, where the presence of large-scale datasets has been foundational for further progress in the field. Many such data benchmarks exist for offline RL as well; a notable one is the D4RL (Fu et al., 2020) benchmark for continuous control which has led to the development of several state-of-the-art offline RL algorithms (Kostrikov et al., 2021; Kumar et al., 2020; Chen et al., 2021) that can scale favorably even in high dimensions. To the best of our knowledge, there are no such existing benchmarks for offline MORL. Even for the online setting, most works in MORL conduct evaluations on toy MDPs (e.g., gridworld) with a few exceptions that include continuous control, e.g., Chen et al. (2019); Xu et al. (2020). This calls for a much-needed push towards more challenging benchmarks for reliable evaluation of MORL, especially in the offline setting.

We introduce Datasets for Multi-Objective Reinforcement Learning (D4MORL), a large-scale benchmark for offline MORL. Our benchmark consists of offline trajectories from 6 multi-objective MuJoCo environments including 5 environments with 2 objectives each (MO-Ant, MO-HalfCheetah, MO-Hopper, MO-Swimmer, MO-Walker2d), and one environment with three objectives (MO-Hopper-3obj). The objectives are conflicting for each environment; for instance, the two objectives in MO-Hopper correspond to jumping and running; in MO-HalfCheetah, MO-Swimmer, and MO-Walker2d, they correspond to the speed and energy savings of the agent. See Appendix A for more details on the semantics of the target objectives for each environment. These environments were first introduced in Xu et al. (2020) for online MORL, and as such, we use their pretrained ensemble policies as building blocks for defining new behavioral policies for dataset collection, which we discuss next.

## 4.1  TRAJECTORY SAMPLING

The quality of the behavioral policy used for sampling trajectories in the offline dataset is a key factor for benchmarking downstream offline RL algorithms. In existing benchmarks for single-objective RL such as D4RL (Fu et al., 2020), the quality of a behavioral policy can be ascertained and varied based on its closeness to a single expert policy, as measured by its scalar-valued returns. For a MOMDP, we do not have the notion of a scalar return and hence, a reference expert policy (or set of policies) should reflect the optimal returns for all possible preferences in the preference space.

We use Prediction-Guided Multi-Objective Reinforcement Learning (PGMORL), a state-of-the-art MORL algorithm for defining reference expert policies. PGMORL (Xu et al., 2020) uses evolutionary algorithms to train an ensemble of policies to approximate the Pareto set. Each reference policy in the ensemble is associated with a unique preference; as for any new preference, it is mapped to the closest preference in the reference set. The number of policies in the ensemble can vary significantly; for instance, we have roughly 70 reference policies for MO-Antand 2445 policies for harder environments such as MO-Hopper-3obj. Given a desired preference, we define two sets of behavioral policies:

1. `Expert` Dataset: We find the best reference policy in the policy ensemble, and always follow the action taken by the selected reference policy.

2. `Amateur` Dataset: As before, we first find the best reference policy in the policy ensemble. With a fixed probability $p$, we randomly perturb the actions of the reference policies. Otherwise, with probability $1 - p$, we take the same action as the reference policy. In D4MORL, we set $p = 0.65$.

Further details are described in Appendix C. In Figure 2, we show the returns of the trajectories rolled out from the expert and amateur policies for the 2 objective environments evaluated for a uniform sampling of preferences. We can see that the expert trajectories typically dominate the amateur trajectories, as desired. For the amateur trajectories, we see more diversity in the empirical returns for both objectives under consideration. The return patterns for the amateur trajectories vary across different environments providing a diverse suite of datasets in our benchmark.

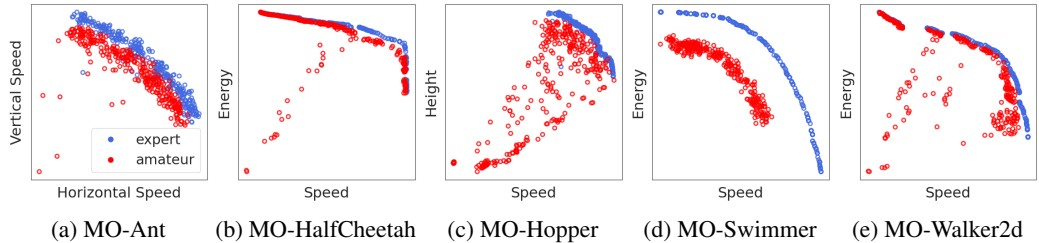

Figure 2: Empirical returns for expert and amateur trajectory datasets for the two-objective environments in D4MORL. For each environment and dataset, we randomly plot returns for 300 trajectories.

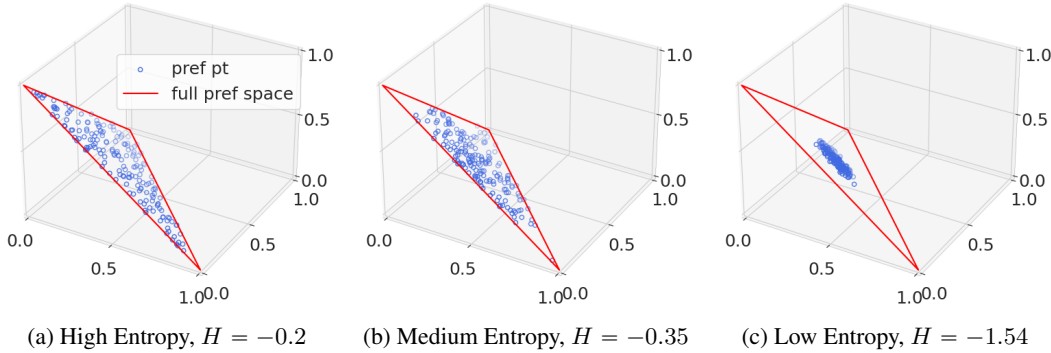

Figure 3: Illustration of the preference distributions for 3 objectives. Entropy is estimated on 50K preference samples using the Vasicek estimator in Scipy (Vasicek, 1976, Virtanen et al., 2020).

## 4.2 PREFERENCE SAMPLING

The coverage of any offline dataset is an important factor in dictating the performance of downstream offline RL algorithms (Levine et al., 2020). For MORL, the coverage depends on both the behavioral MORL policy as well as the distribution of preferences over which this policy is evaluated. We use the following protocols for sampling from the preference space $\Omega$. First, we restrict our samples to lie within a physically plausible preference space $\Omega^* \subseteq \Omega$ covered by the behavioral policy $\pi_\beta$. For instance, MO-Hopper has two objectives: jumping and running. Since the agent can never gain running rewards without leaving the floor. Thus, the preference of 100% running and 0% jumping is not achievable and excluded from our preference sampling distribution.

Second, we are primarily interested in offline trajectories that emphasize competition between multiple objectives rather than focusing on a singular objective. To enforce this criterion, we define 3 sampling distributions concentrated around the centroid of the preference simplex. The largest spread distribution samples uniformly from $\Omega^*$ and is denoted as **High-Entropy** (High-H). Next, we have a **Medium-Entropy** (Med-H) distribution specified via samples of Dirichlet distributions with large values of their concentration hyperparameters (aka $\alpha$). Finally, we have a **Low-Entropy** (Low-H) distribution that is again specified via samples of Dirichlet distributions but with low values of their concentration hyperparameters. We illustrate the samples for each of the preference distributions along with their empirical entropies in Figure 3. Further details on the sampling distributions are deferred to Appendix B. By ensuring different levels of coverage, we can test the generalizability of an MORL policy to preferences unseen during training. In general, we expect Low-H to be the hardest of the three distributions due to its restricted coverage, followed by Med-H and High-H.

**Overall Data Generation Pipeline.** The pseudocode for generating the dataset is described in Algorithm 1. Given a preference distribution, we first sample a preference $\omega$ and query the closest behavioral policy in either the amateur/expert ensemble matching $\omega$. We roll out this policy for $T$ time steps (or until the end of an episode if sooner) and record the state, action, and reward information. Each trajectory in our dataset is represented as:

$$\tau = <\omega, s_1, a_1, r_1, \ldots, s_T, a_T, r_T>$$

---

**Algorithm 1** Data Collection in D4MORL

---

  **procedure** COLLECT(prefDist, nTraj, env, pretrainedAgents, T)
    agents = pretrainedAgents
    prefs = prefDist(nTraj)
    all_trajs = []
    **for** $\omega$ in prefs **do**
      agent = closestAgent(agents, $\omega$)
      $s$ = env.reset()
      done = False
      $\tau = [\omega]$
      t = 0
      **while** (NOT done) AND (t < T) **do**
        $a$ = agent.get_action($s$)
        $s'$, done, $r$ = env.step($a$)
        append $s$, $a$, $s'$, $r$ to $\tau$
        $s = s'$
        t = t + 1
      append $\tau$ to all_trajs
  **return** all_trajs

---

For every environment in D4MORL, we collect 50K trajectories of length $T = 500$ for both expert and amateur trajectory distributions under each of the 3 preference distributions. Overall, this results in a total of 1.8M trajectories over all 6 environments, which corresponds to roughly 867M time steps. We refer the reader to Table 5 in Appendix B for additional statistics on the dataset.

## 5 PARETO-EFFICIENT DECISION AGENTS (PEDA)

In this section, we propose *Pareto-Efficient Decision Agents (PEDA)*, a new family of offline multi-objective RL agents. PEDA aims to achieve Pareto-efficiency by extending Decision Transformers (Chen et al., 2021) into multi-objective setting. We first introduce the architecture of Decision Transformers (DT) and its variant, Reinforcement Learning Via Supervised Learning (RvS), followed by our modifications extending them to the multi-objective setting.

DT casts offline RL as a conditional sequence modeling problem that predicts the next action by conditioning a transformer on past states, actions, and desired returns. The desired returns are defined as returns-to-go (RTG) $g_t = \sum_{t'=t}^{T} r_{t'}$, the future returns that this action is intended to achieve. Therefore, the trajectory is represented by $\tau = <s_1, a_1, g_1, \ldots, s_T, a_T, g_T>$. In practice, we use a causally masked transformer architecture such as GPT (Radford et al., 2019) to process this sequence and predict the actions by observing the past $K$ timesteps consisting of $3K$ tokens. DT and its variants have been shown to be more stable and robust to optimize due to the simplicity of loss function; easier to scale to more complex settings such as environments with high-dimensional actions or states, and agents with broad capabilities such as multitask settings (Lee et al., 2022). Hence, we adopt Decision Transformers (Chen et al., 2021) as the representative base algorithm on which we build our work.

In follow-up work, Emmons et al. (2021) extend DT and shows that even multi-layer perceptrons conditioned on the average returns-to-go can achieve similar performance without the use of transformers. They call their model Reinforcement Learning Via Supervised Learning (RvS). However, RvS is generally not very stable when conditioned on very large returns, unlike DT.

### 5.1 MULTI-OBJECTIVE REINFORCEMENT LEARNING VIA SUPERVISED LEARNING

In PEDA, our goal is to train a single preference-conditioned agent for offline MORL. By including preference conditioning, we enable the policy to be trained on arbitrary offline data, including trajectories collected from behavioral policies that are associated with alternate preferences. To parameterize our policy agents, we extend the DT and RvS architectures to include preference tokens and vector-valued returns. We refer to such preference-conditioned extensions of these architectures as MODT(P) and MORvS(P) respectively, which we describe next.

**Preference Conditioning.** Naively, we can easily incorporate the preference $\omega$ into DT by adding this token for each timestep and feeding it a separate embedding layer. However, empirically we find that such a model design tends to ignore $\omega$ and the correlation between the preferences and predicted actions is weak. Therefore, we propose to concatenate $\omega$ to other tokens before any layers in MODT(P). Concretely, we define $s^* = s \oplus \omega$, $a^* = a \oplus \omega$, and $g^* = g \oplus \omega$ where $\oplus$ denotes the concatenation operator. Hence, triples of $s^*$, $a^*$, $g^*$ form the new trajectory. As for MORvS(P), we concatenate the preference with the states and the average RTGs by default and the network interprets everything as one single input.

**Multi-Objective Returns-to-Go.** Similar to RTG for the single objective case, we can define vector-valued RTG as $g_t = \sum_{t'=t}^T r_{t'}$ Given a preference vector $\omega$, we can scalarize the total returns-to-go as $\hat{g}_t = \omega^T g_t$. In principle, the scalarized RTG $\hat{g}_t$ can be recovered given the preference vector $\omega$ and the vector-valued RTG $g_t$. However, empirically we find that directly feeding MODT/MORvS with the preference-weighted RTG vector $g_t \odot \omega$ is slightly preferable for stable training, where $\odot$ denotes the elementwise product operator.

Another unique challenge in the MORL setting concerns the scale of different objectives. Since different objectives can signify different physical quantities (e.g., energy and speed), the choice of scaling can influence policy optimization. We adopt a simple normalization scheme, where the returns for each objective are normalized by subtracting the minimum observed value for that objective and dividing it by the range of values (max-min). Note that the maximum and minimum are computed based on the offline dataset and hence, they are not necessarily the true min/max objective values. Post this normalization, the values for every objective in the trajectory are on the same scale between 0 and 1. For evaluating the hypervolume and sparsity, we use the unnormalized values so that we can make comparisons across different datasets that may have different min/max boundaries.

**Training.** We follow a simple supervised training procedure where we train the policies on randomly sampled mini-batches with MSE loss (for continuous actions). In MODT and MODT(P), the input states, actions, and returns-to-go (with concatenated preferences) are treated as tokens and embedded through one layer of MLP. We apply a layer of MLP and Tanh on the last hidden state of GPT-2 transformer to predict next action. In MORvS and MORvS(P), we use only information from the current timestep and MLP layers to predict the next action.

# 6 EXPERIMENTS

In this section, we evaluate the performance of PEDA on D4MORL benchmark. First, we investigate the benefits of preference conditioning by evaluating on decision transformers (DT) and RvS (MORvS) where no preference information is available and we scalarize multi-objective vector returns into weighted sums. We denote our methods with preference conditioning as MODT(P) and MORvS(P). Second, we compare our methods with classic imitation learning and temporal difference learning algorithms with preference conditioning.

**Imitation learning.** Imitation learning simply uses supervised loss to train a mapping from states (w/ or w/o concatenating preferences) to actions. We use behavioral cloning (BC) here and train multi-layer MLPs as models named BC (w/o preference) and BC(P) (w/ preference).

**Temporal difference learning.** Conservative Q-Learning (CQL) (Kumar et al., 2020) is the state-of-the-art standard offline RL method, which learns a conservative Q-function $f : \mathcal{S} \times \mathcal{A} \to \mathbb{R}$ through neural networks. We modify the network architecture such that it also takes preference vectors as inputs to learn a preference-conditioned Q-function $f^* : \mathcal{S} \times \mathcal{A} \times \Omega \to \mathbb{R}$. We denote this method as CQL(P).

## 6.1 MULTI-OBJECTIVE OFFLINE BENCHMARK

Table 1: Hypervolume performance on `High-H-Expert` dataset. PEDA variants MODT(P) and MORvS(P) always approach the expert behavioral policy. (B: Behavioral policy)

| Environments | B | MODT(P) | MORvS(P) | BC(P) | CQL(P) | MODT | MORvS | BC | CQL |
|---|---|---|---|---|---|---|---|---|---|
| MO-Ant ($10^6$) | 6.32 | 6.21±.01 | **6.41±.01** | 4.88±.17 | 5.76±.10 | 5.52±.16 | 5.52±.02 | 0.84±.60 | 3.52±.45 |
| MO-HalfCheetah ($10^6$) | 5.79 | 5.73±.00 | **5.78±.00** | 5.54±.05 | 5.63±.04 | 5.59±.03 | 4.19±.74 | 1.53±.09 | 3.78±.46 |
| MO-Hopper ($10^7$) | 2.09 | **2.00±.02** | **2.02±.02** | 1.23±.10 | 0.33±.39 | 1.68±.03 | 1.73±.07 | 0.28±.21 | 0.02±.02 |
| MO-Hopper-3obj ($10^{10}$) | 3.73 | **3.38±.05** | **3.42±.10** | 2.29±.07 | 0.78±.24 | 1.05±.43 | 2.53±.06 | 0.06±.02 | 0.00±.00 |
| MO-Swimmer ($10^4$) | 3.25 | 3.15±.02 | **3.24±.00** | 3.21±.00 | **3.22±.08** | 2.49±.19 | 3.19±.01 | 1.68±.38 | 2.08±.08 |
| MO-Walker2d ($10^6$) | 5.21 | 4.89±.05 | **5.14±.01** | 3.74±.11 | 3.21±.32 | 0.65±.46 | 5.10±.02 | 0.07±.02 | 0.82±.62 |

**Hypervolume.** We compare hypervolume of our methods with all baselines on expert datasets in Table 1 as well as amateur dataset in Table 2. For the two-objective environments, we evaluate the

Table 2: Hypervolume performance on `High-H-Amateur` dataset. PEDA variants still approach or even exceed the behavioral policy even when a considerable portion of data is suboptimal. MODT(P) and MORvS(P) still present to be the strongest models and outperform other baselines. (B: Behavioral policy)

| Environments | B | MODT(P) | MORvS(P) | BC(P) | CQL(P) | MODT | MORvS | BC | CQL |
|---|---|---|---|---|---|---|---|---|---|
| MO-Ant ($10^6$) | 5.61 | 5.92±.04 | **6.07±.02** | 4.37±.06 | 5.62±.23 | 4.88±.60 | 4.37±.56 | 2.34±.15 | 2.80±.68 |
| MO-HalfCheetah ($10^6$) | 5.68 | 5.69±.01 | **5.77±.00** | 5.46±.02 | 5.54±.02 | 5.51±.01 | 4.66±.05 | 2.92±.38 | 4.41±.08 |
| MO-Hopper ($10^7$) | 1.97 | **1.81±.05** | 1.76±.03 | 1.35±.03 | 1.64±.01 | 1.54±.08 | 1.57±.01 | 0.01±.01 | 0.00±.06 |
| MO-Hopper-3obj ($10^{10}$) | 3.09 | 1.04±.16 | **2.77±.24** | 2.42±.18 | 0.59±.42 | 1.64±.23 | 1.30±.22 | 0.03±.01 | 0.10±.16 |
| MO-Swimmer (1) | 2.11 | 1.67±.22 | 2.79±.03 | 2.82±.04 | 1.69±.93 | 0.96±.19 | **2.93±.03** | 0.46±.15 | 0.74±.47 |
| MO-Walker2d ($10^4$) | 4.99 | 3.10±.34 | **4.98±.01** | 3.42±.42 | 1.78±.33 | 3.76±.34 | 4.32±.05 | 0.91±.36 | 0.76±.81 |

Table 3: Sparsity (↓) performance on `High-H-Expert` dataset. MODT(P) and MORvS(P) have a lower density. BC(P) also has a competitive sparsity in smaller environments such as Swimmer.

| Environments | MODT(P) | MORvS(P) | BC(P) | CQL(P) |
|---|---|---|---|---|
| MO-Ant ($\times 10^4$) | 8.26±2.22 | 6.50±0.81 | 46.2±16.4 | **0.58±0.10** |
| MO-HalfCheetah ($\times 10^4$) | 1.24±0.23 | 0.67±0.05 | 1.78±0.39 | **0.10±0.00** |
| MO-Hopper ($\times 10^5$) | 16.3±10.6 | **3.03±0.36** | 52.5±4.88 | 2.84±2.46 |
| MO-Hopper-3obj ($\times 10^5$) | 1.40±0.44 | 2.72±1.93 | **0.72±0.09** | 2.60±3.14 |
| MO-Swimmer ($\times 1$) | 15.0±7.49 | **4.39±0.07** | 4.50±0.39 | 13.6±5.31 |
| MO-Walker2d ($\times 10^4$) | **0.99±0.44** | 3.22±0.73 | 75.6±52.3 | 6.23±10.7 |

models on 501 equally spaced preference points in the range [0, 1]; on the three-objective environment MO-Hopper-3obj, models are evaluated on 325 equally spaced points. Each point is evaluated 5 times with random environment re-initialization, and the median value is recorded. Finally, all the results are based on 3 random seeds and we report the mean performance along with the standard error. In Table 1 and Table 2, we can see that MODT(P) and MORvS(P) outperform other baselines and has a relatively very low standard error. Also, PEDA variants including MODT(P) and MORvS(P) approaches the behavioral policy upper-bound.

**Sparsity.** We also evaluate sparsity performance. Since sparsity comparison is only meaningful between models that are sensitive to preference and have a relatively similar hypervolume performance, we only show results for models that concatenate preference. Overall, MORvS(P) has the lowest sparsity in most environments, while at the same time featuring an outstanding hypervolume.

## 6.2 ABLATION STUDY

**Pareto front approximation.** We ablate how well the MODT(P) and MORvS(P) can approximate the Pareto front through conditioning on different preference points. We show the results in Figure 4, where we can see that the models can approximate the Pareto front while having some dominated points colored in pink mostly in the MO-Hopper and MO-Walker2d environments. The results are based on the average of 3 seeds, and the full plot can be found in Appendix G.

Table 4: Sparsity (↓) performance on `High-H-Amateur` dataset. We can see that all models still have a similar or stronger sparsity performance when trained on amateur datasets. Furthermore, MORvS(P) still presents the strongest performance. While BC(P) has strong performance in MO-Hopper-3obj and MO-Swimmer, it also fails to give a dense solution in other environments and has a higher standard error.

| Environments | MODT(P) | MORvS(P) | BC(P) | CQL(P) |
|---|---|---|---|---|
| MO-Ant ($\times 10^4$) | 8.72±.77 | 5.24±.52 | 25.9±16.4 | **1.06±.28** |
| MO-HalfCheetah ($\times 10^4$) | 1.16±.42 | **0.57±.09** | 2.22±.91 | **0.45±.27** |
| MO-Hopper ($\times 10^5$) | **1.61±.29** | 3.50±1.54 | 2.42±1.08 | 3.30±5.25 |
| MO-Hopper-3obj ($\times 10^5$) | 10.23±2.78 | **1.03±.11** | 0.87±.29 | 2.00±1.72 |
| MO-Swimmer ($\times 1$) | 2.87±1.32 | 1.03±.20 | 5.05±1.82 | 8.87±6.24 |
| MO-Walker2d ($\times 10^4$) | 164.2±13.5 | **1.94±.06** | 53.1±34.6 | 7.33±5.89 |

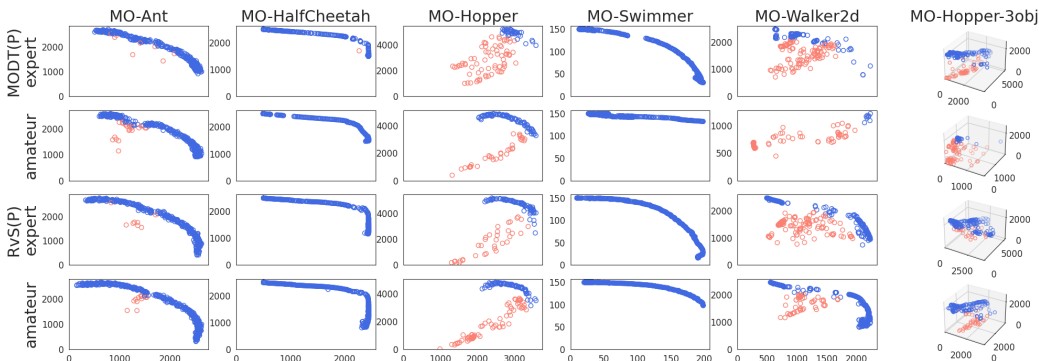

Figure 4: We show that MODT(P) and MORvS(P) can be good approximator to the Pareto front. There are relatively more dominated points in MO-Hopper and MO-Walker2d colored in red.

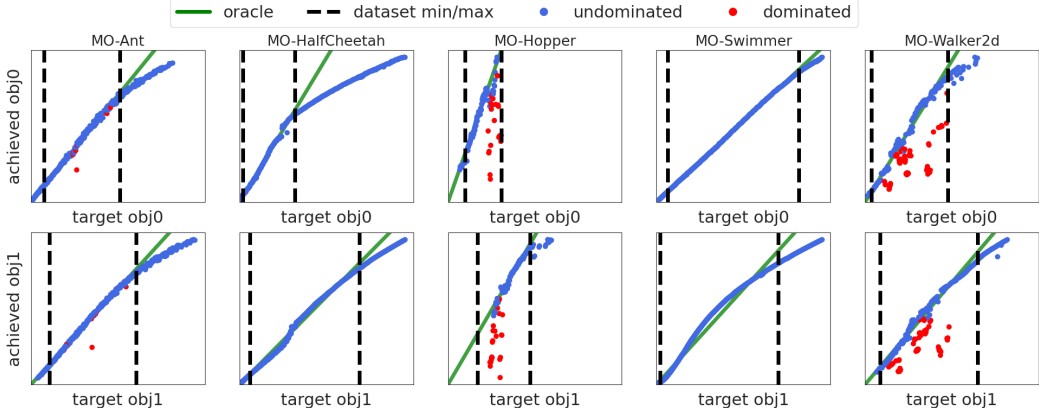

Figure 5: We show that MORvS(P) Model can follow the given target reward that is within the dataset's minimum and maximum record. The plots are for all the two-objective environments. In addition, MO-Hopper and MO-Walker2d present to be the most challenging environments for PEDA variants, featuring more dominated solutions than other environments.

**Return distribution.** We ablate how well MODT(P) and MORvS(P) follow their given target return, based on a normalized and weighted value. We present the results in Figure 5 for MORvS(P) under `High-H-Expert` datasets and refer to Appendix H for full settings. Here, we see that the models follow the oracle line nicely when conditioned on the target within the dataset distribution, and generalize to targets outside of the dataset distribution as well.

## 7 CONCLUSION

We proposed a new problem setup for offline Multi-Objective Reinforcement Learning to scale Pareto-Efficient decision-making using offline datasets. To characterize progress, we introduced D4MORL, a dataset benchmark consisting of offline datasets generated from behavioral policies of different fidelities (expert/amateur) and rolled out under preference distributions with varying entropies (high/medium/low). Then, we propose PEDA, a family of offline MORL policy optimization algorithms based on decision transformers. To our knowledge, the PEDA variants are the first offline MORL policies that support continuous action and preference spaces. We showed that by concatenating and embedding preferences together with other inputs, our policies can effectively approximate the Pareto front of the underlying behavioral policy as measured by the hypervolume and sparsity metrics. Our proposed family includes MLP and transformer-based variants, viz. the MORvS(P) and MODT(P), with MORvS(P) performing the best overall. In some scenarios, the learned policies can also generalize to higher target rewards that exceed the data distribution.

## REPRODUCIBILITY STATEMENT

Our code is available at: `https://github.com/baitingzbt/PEDA`.

## ACKNOWLEDGEMENTS

AG's research is supported by a Meta Research Award and a Cisco grant.

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

# A    ENVIRONMENT DESCRIPTION

All environments are the same as in Xu et al., 2020, except for when resetting the environment, each parameter is uniformly sampled from the $[x - 10^{-3}, x + 10^{-3}]$ with $x$ being the default value. Except we always reset *height* as 1.25 for MO-Hopper and MO-Hopper-3objsince this parameter directly relates to the reward function. All environments have a max episode length of 500 steps per trajectory, but the agent may also die before reaching the maximum length.

## A.1    MO-ANT

The two objectives in MO-Ant are achieved distance in x and y axes respectively, denoted as $\boldsymbol{r} = [r_t^{vx}, r_t^{vy}]^{\intercal}$.

Consider the position of the agent is represented as $(x_t, y_t)$ at time $t$ and takes the action $a_t$. The agent has a fixed survival reward $r^s = 1.0$, $dt = 0.05$, and an action cost of $r^a = \frac{1}{2} \sum_k a_k^2$. The rewards are calculated as:

$$
\begin{aligned}
r_t^{vx} &= (x_t - x_{t-1}) \,/\, dt + r^s - r^a \\
r_t^{vy} &= (y_t - y_{t-1}) \,/\, dt + r^s - r^a
\end{aligned}
\tag{1}
$$

## A.2    MO-HALFCHEETAH

The two objectives in MO-HalfCheetah are running speed, and energy saving, denoted as $\boldsymbol{r} = [r_t^v, r_t^e]^{\intercal}$.

Consider the position of the agent is represented as $(x_t, y_t)$ at time $t$ and takes the action $a_t$. The agent has a fixed survival reward $r^s = 1.0$, fixed $dt = 0.05$, and an action cost of $r^a = \sum_k a_k^2$. The rewards are calculated as:

$$
\begin{aligned}
r_t^v &= \min\{4.0, \ (x_t - x_{t-1}) \,/\, dt\} + r^s \\
r_t^e &= 4.0 - r^a + r^s
\end{aligned}
\tag{2}
$$

## A.3    MO-HOPPER

The two objectives in MO-Hopper are running and jumping, denoted as $\boldsymbol{r} = [r^r, r^j]^{\intercal}$.

Consider the position of the agent is represented as $(x_t, h_t)$ at time $t$ and takes the action $a_t$. The agent has a fixed survival reward $r^s = 1.0$, a fixed initial height as $h_{init} = 1.25$, a fixed $dt = 0.01$, and an action cost of $r^a = 2 \times 10^{-4} \sum_k a_k^2$. The rewards are calculated as:

$$
\begin{aligned}
r_t^r &= 1.5 \times (x_t - x_{t-1}) \,/\, dt + r^s - r^a \\
r_t^j &= 12 \times (h_t - h_{init}) \,/\, dt + r^s - r^a
\end{aligned}
\tag{3}
$$

## A.4    MO-HOPPER-3OBJ

The physical dynamics are the same in MO-Hopper and MO-Hopper-3obj, while this environment has 3 objectives: running, jumping, and energy saving. The rewards are denoted as $\boldsymbol{r} = [r^r, r^j, r^e]^{\intercal}$.

Consider the position of the agent is represented as $(x_t, h_t)$ at time $t$ and takes the action $a_t$. The agent has a fixed survival reward $r^s = 1.0$, a fixed initial height as $h_{init} = 1.25$, a fixed $dt = 0.01$, and an action cost of $r^a = \sum_k a_k^2$. The rewards are calculated as:

$$
\begin{aligned}
r_t^r &= 1.5 \times (x_t - x_{t-1}) \,/\, dt + r^s \\
r_t^j &= 12 \times (h_t - h_{init}) \,/\, dt + r^s \\
r_t^e &= 4.0 - r^a + r^s
\end{aligned}
\tag{4}
$$

## A.5    MO-SWIMMER

The two objectives in MO-Swimmer are speed and energy saving, denoted as $\boldsymbol{r} = [r^v, r^e]^\intercal$.

Consider the position of the agent is represented as $(x_t, y_t)$ at time $t$ and takes the action $a_t$. The agent has a fixed $dt = 0.05$, and an action cost of $r^a = \sum_k a_k^2$. The rewards are calculated as:

$$
\begin{aligned}
r_t^v &= (x_t - x_{t-1}) \,/\, dt \\
r_t^e &= 0.3 - 0.15 \times r^a
\end{aligned}
\tag{5}
$$

## A.6    MO-WALKER2D

The objectives in MO-Walker2d are speed and energy saving, denoted as $\boldsymbol{r} = [r^v, r^e]^\intercal$.

Consider the position of the agent is represented as $(x_t, y_t)$ at time $t$ and takes the action $a_t$. The agent has a fixed survival reward $r^s = 1.0$, a fixed $dt = 0.008$, and an action cost of $r^a = \sum_k a_k^2$. The rewards are calculated as:

$$
\begin{aligned}
r_t^v &= (x_t - x_{t-1}) \,/\, dt + r^s \\
r_t^e &= 4.0 - r^a + r^s
\end{aligned}
\tag{6}
$$

To uniformly sample the `High-H` data from the entire preference space, the problem is equivalent to sampling from a $n$-dimensional simplex, where $n$ is the number of objectives. The resulting sampling is:

$$
\boldsymbol{\omega}^{\text{high}} \sim ||\boldsymbol{f}_{\exp}(\,\cdot\,, \lambda = 1)||_1
\tag{7}
$$

We take the 1-norm following the exponential distribution to make sure each preference add up to 1. When $\boldsymbol{\Omega}^* \neq \boldsymbol{\Omega}$, we perform rejection sampling to restrict the range.

To sample the `Med-H` and `Low-H` data, we first sample $\boldsymbol{\alpha}$ from a non-negative uniform distribution, then sample the corresponding Dirichlet preference. Here, we sample a *different* alpha to make sure the center of the Dirichlet changes and thus allows more variation.

$$
\begin{aligned}
\boldsymbol{\omega}^{\text{med}} &\sim \boldsymbol{f}_{\text{Dirichlet}}(\boldsymbol{\alpha}) \; ; \text{ where } \boldsymbol{\alpha} \sim \text{Unif}(0, 10^6) \\
\boldsymbol{\omega}^{\text{low}} &\sim \boldsymbol{f}_{\text{Dirichlet}}(\boldsymbol{\alpha}) \; ; \text{ where } \boldsymbol{\alpha} \sim \text{Unif}(1/3 \times 10^6, 2/3 \times 10^6)
\end{aligned}
\tag{8}
$$

For sampling from behavioral policy consists of a group of single-objective policies $\pi_\beta = \{\pi_1, \ldots, \pi_B\}$ with $B$ being the total number of candidate policies, we recommend first find the expected unweighted raw rewards $\boldsymbol{G}^{\pi_1}, \ldots, \boldsymbol{G}^{\pi_B}$. Then, find the estimated $\hat{\boldsymbol{\omega}}^{\pi_1}, \ldots, \hat{\boldsymbol{\omega}}^{\pi_B}$ by letting $\hat{\omega}_i^{\pi_b} = \frac{G_i^{\pi_b}}{\sum_{j=1}^n G_j^{\pi_b}}$, which represents the estimated preference on $i^{\text{th}}$ objective of $b^{\text{th}}$ candidate policy. For a sampled preference $\boldsymbol{\omega} \sim \boldsymbol{\Omega}^*$, use the policy that provides the smallest euclidean distance $d(\boldsymbol{\omega}, \hat{\boldsymbol{\omega}}^{\pi_b})$. Empirically, this means picking the candidate policy that has the expected reward ratio closest to $\boldsymbol{\omega}$.

## B    DATASET DETAILS

To uniformly sample the `High-H` data from the entire preference space, we can equivalently sample from a $n$-dimensional simplex, where $n$ is the number of objectives. The resulting sampling scheme is:

$$
\boldsymbol{\omega}^{\text{high}} \sim ||\boldsymbol{f}_{\exp}(\,\cdot\,, \lambda = 1)||_1
\tag{9}
$$

The 1-norm following the exponential distribution makes sure each preference vector have entries add up to 1. When $\boldsymbol{\Omega}^* \neq \boldsymbol{\Omega}$, we perform rejection sampling to restrict the range.

Table 5: A comprehensive view of the dataset. All datasets have a 500 maximum step per trajectory, and 50K trajectories are collected under each setting. As indicated by average step per trajectory, we can see `Amateur` trajectories are always shorter or same as `Expert`, thus leading to a lower return.

|  | max step per traj | expert avg. step per traj | amateur avg. step per traj | trajectories per dataset |
|---|---|---|---|---|
| MO-Ant | 500 | 500 | 500 | 50K |
| MO-HalfCheetah | 500 | 499.91 | 482.11 | 50K |
| MO-Hopper | 500 | 499.94 | 387.72 | 50K |
| MO-Hopper-3obj | 500 | 499.99 | 442.87 | 50K |
| MO-Swimmer | 500 | 500 | 500 | 50K |
| MO-Walker2d | 500 | 500 | 466.18 | 50K |

To sample the `Med-H` and `Low-H` data, we first sample $\alpha$ from a non-negative uniform distribution, then sample the corresponding Dirichlet preference. Here, we sample a *different* alpha to make sure the mode of the Dirichlet changes and thus allows more variation.

$$\begin{aligned}
\boldsymbol{\omega}^{\text{med}} &\sim \boldsymbol{f}_{\text{Dirichlet}}(\boldsymbol{\alpha}) \text{ ; where } \boldsymbol{\alpha} \sim \text{Unif}(0, 10^6) \\
\boldsymbol{\omega}^{\text{low}} &\sim \boldsymbol{f}_{\text{Dirichlet}}(\boldsymbol{\alpha}) \text{ ; where } \boldsymbol{\alpha} \sim \text{Unif}(1/3 \times 10^6, 2/3 \times 10^6)
\end{aligned} \tag{10}$$

Since our behavioral policy is consists of a group of single-objective policies $\pi_\beta = \{\pi_1, \ldots, \pi_B\}$ with $B$ being the total number of candidate policies, we first find the expected unweighted raw rewards $\boldsymbol{G}^{\pi_1}, \ldots, \boldsymbol{G}^{\pi_B}$. Then, we find the estimated preferences $\hat{\boldsymbol{\omega}}^{\pi_1}, \ldots, \hat{\boldsymbol{\omega}}^{\pi_B}$ by letting $\hat{\omega}_i^{\pi_b} = \frac{G_i^{\pi_b}}{\sum_{j=1}^{n} G_j^{\pi_b}}$ on $i^{\text{th}}$ objective of $b^{\text{th}}$ candidate policy. For each sampled preference $\boldsymbol{\omega} \sim \boldsymbol{\Omega^*}$ following (9) or (10), we sample a complete trajectory using the single-objective behavioral policy that provides the smallest euclidean distance $\min d(\boldsymbol{\omega}, \hat{\boldsymbol{\omega}}^{\pi_b})$. Empirically, this means picking the candidate policy that has the expected reward ratio closest to $\boldsymbol{\omega}$.

## C  EXPERT & AMATEUR DATASETS

**In `Expert` collection**, we sample trajectories using the fully-trained behavioral policy $\pi_\beta$. In this paper, we use PGMORL by Xu et al., 2020 as our behavioral policy $\pi_\beta$

$$\boldsymbol{a}_{t+1}^{\text{expert}} = \pi_\beta(\boldsymbol{a}|\boldsymbol{s} = \boldsymbol{s_t}, \boldsymbol{\omega} = \boldsymbol{\omega_t}) \tag{11}$$

**In the `Amateur` collection**, the policies has a 35% chance being stochastic on top of the expert collection. Actions has a chance being stochastic, during which it is scaled from the expert action, as following:

$$\boldsymbol{a}_{t+1}^{\text{amateur}} = \begin{cases} \boldsymbol{a}_{t+1}^{\text{expert}} & 35\% \\ \boldsymbol{a}_{t+1}^{\text{expert}} \times \text{Unif}(0.35, 1.65) & 65\% \end{cases} \tag{12}$$

In the MO-Swimmer environment only, we let actions has a 35% chance to be a uniform random sample from the entire action space rather than being the same as expert to increase variance and achieve a performance similar to amateur. The resulting strategy for MO-Swimmer is:

$$\boldsymbol{a}_{t+1}^{\text{amateur}} = \begin{cases} \text{Unif}(\mathcal{A}) & 35\% \\ \boldsymbol{a}_{t+1}^{\text{expert}} \times \text{Unif}(0.35, 1.65) & 65\% \end{cases} \tag{13}$$

## D  FINDING APPROPRIATE MULTI-OBJECTIVE RTG

In Decision Transformer Chen et al. (2021) and Emmons et al. (2021), RTG denotes the future desired reward. In MORL, however, designing appropriate *multi-objective* RTG is necessary. On top

of discounting each objective's desired reward separately, we empirically find that since some objectives are inherently conflicting, setting RTG high for one objective means we should accordingly lower RTG for other objectives (i.e. we shouldn't use maximum RTG for both). In this way, our test-time RTG can follow closer to the training distribution.

In this paper, we use linear regression $G = f(\omega)$ to find corresponding RTG conditioned on the given preference. Figure 6 demonstrates the weighted RTG of the "running" objective as a function of its preference in MO-Hopper where the conflicting objectives are "running" and "jumping". It is clear that RTG closely correlates with the conditioned preference for running and we should adjust the initial RTG during test-time accordingly.

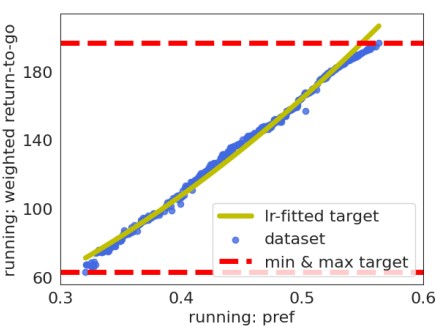

Finally, we only use the learned linear regression model from `Expert` dataset. This is because regression models fitted on sub-optimal data can easily produce an RTG lower than optimal. In practice, we can easily achieve a similar result by training the regression model only on the best-performing trajectories for respective preferences. Other regression or clustering methods to find appropriate RTG can also work, and we leave it as future works especially when not assuming linearly weighted objectives.

Figure 6: RTG fitted through linear regression (lr) models stay closer to the dataset distribution. Furthermore, linear regression models can also efficiently generalize to unseen preferences during test time.

## E  TRAINING DETAILS

In this section, we list our hyper-parameters and model details. In specific, we use the same hyper-parameters for all algorithms, except for the learning rate scheduler and warm-up steps. In MODT family, inputs are first embedded by a 1-layer fully-connected network, and n_layer represents the number of transformer blocks; in BC family, n_layer represents the number of MLP layers to embed each input; in MORvS and MORvS(P), we leverage the same embedding strategy in Emmons et al. (2021). Additionally, we consider MORvS and MORvS(P) both have context length of 1 because they only use the current state to predict the next action, whereas MODT and BC use the past 20.

### E.1  PARAMETERS

| Hyperparameter | MODT | MORvS | BC |
|---|---|---|---|
| Context Length - K | 20 | 1 | 20 |
| Batch Size | | 64 | |
| Hidden Size | | 512 | |
| Learning Rate | | 1e-4 | |
| Weight Decay | | 1e-3 | |
| Dropout | | 0.1 | |
| n_layer | | 3 | |
| Optimizer | | AdamW | |
| Loss Function | | MSE | |
| LR Scheduler | lambda | None | lambda |
| Warm-up Steps | 10000 | N/A | 4000 |
| Activation | | ReLU | |

### E.2 TRAINING STEPS

| Dataset Name | MODT Steps | RvS/BC Steps |
|---|---|---|
| MO-Ant | 20K | 200K |
| MO-HalfCheetah | 80K | 200K |
| MO-Hopper | 400K | 200K |
| MO-Hopper-3obj | 400K | 200K |
| MO-Swimmer | 260K | 200K |
| MO-Walker2d | 360K | 200K |

### E.3 OTHER ATTEMPTED ARCHITECTURES FOR MODT AND MODT(P)

We tried the following MODT architectures in our preliminary experiments. We picked *Case 4* eventually as it gave the best performance in our experiments.

1. Consider $\omega$ as an independent token of the transformer.

2. Train a separate embedding for $\omega$, concatenate the embeddings to get $f_{\phi_s}(s) \bigoplus f_{\phi_\omega}(\omega)$, $f_{\phi_a}(a) \bigoplus f_{\phi_\omega}(\omega)$, and $f_{\phi_g}(g) \bigoplus f_{\phi_\omega}(\omega)$ then pass into the transformer.

3. Add another MLP layer on top of *Case 2* after concatenation, then pass output into the transformer.

4. Concatenate $\omega$ to other tokens before any layers. This means we have $s^* = s \bigoplus \omega$, $a^* = a \bigoplus \omega$, and $g^* = g \bigoplus \omega$.

## F OTHER EVALUATION METRICS

Among a variety of metrics for MORL, we use Hypervolume (HV) and Sparsity (SP) to benchmark models for several reasons. First, many metrics such as the $\epsilon$-metric require prior knowledge of the *true* Pareto Fronts, which are not available for our MuJoCo Environments. Second, we only assume linear reward function and cannot collect real-time user feedback, thus utility-based metrics such as *expected utility metric* (EUM) are not applicable. Finally, using the same metric as in the original behavioral policy paper facilitate algorithm comparisons.

## G PARETO SET VISUALIZATIONS

We present the Pareto Set visualizations for all of our models trained under each `High-H` dataset in Figure 7. Each point in each subplot is based on the average result of 3 seeds. In 2 objective environments, we evaluate the model using 501 equally spaced preference points. In 3 objective environments, we use 351 equally spaced preference points instead. Since the environments are stochastically initialized, we evaluate 5 times at each preference point and take the mean value. This makes each point the average value of 15 runs. We here allow a small tolerance for coloring the dominated points.

If a preference point is within the achievable preference $\Omega^*$ but the solution is dominated, we color it in red. Since our models are conditioned on continuous preference points and environments are initialized stochastically, we give a small tolerance (3%-8%) for points to be colored in blue. The hypervolume and sparsity metrics, on the other hand, are based on strictly undominated solutions without tolerance.

## H MEDIUM & LOW ENTROPY DATASET TRAINING

We train on the Medium-Entropy and Low-Entropy datasets for the MO-HalfCheetah environment. Overall, models have a similar performance under `Med-H` and `High-H` datasets but suffer when only trained on `Low-H`. We present the results in Table 6, in which we illustrate that the `Low-H` dataset has a worse expert and amateur performance due to reduced variability on preference. However, MODT(P) and MORvS(P) are still able to get close to or exceed in hypervolume on all

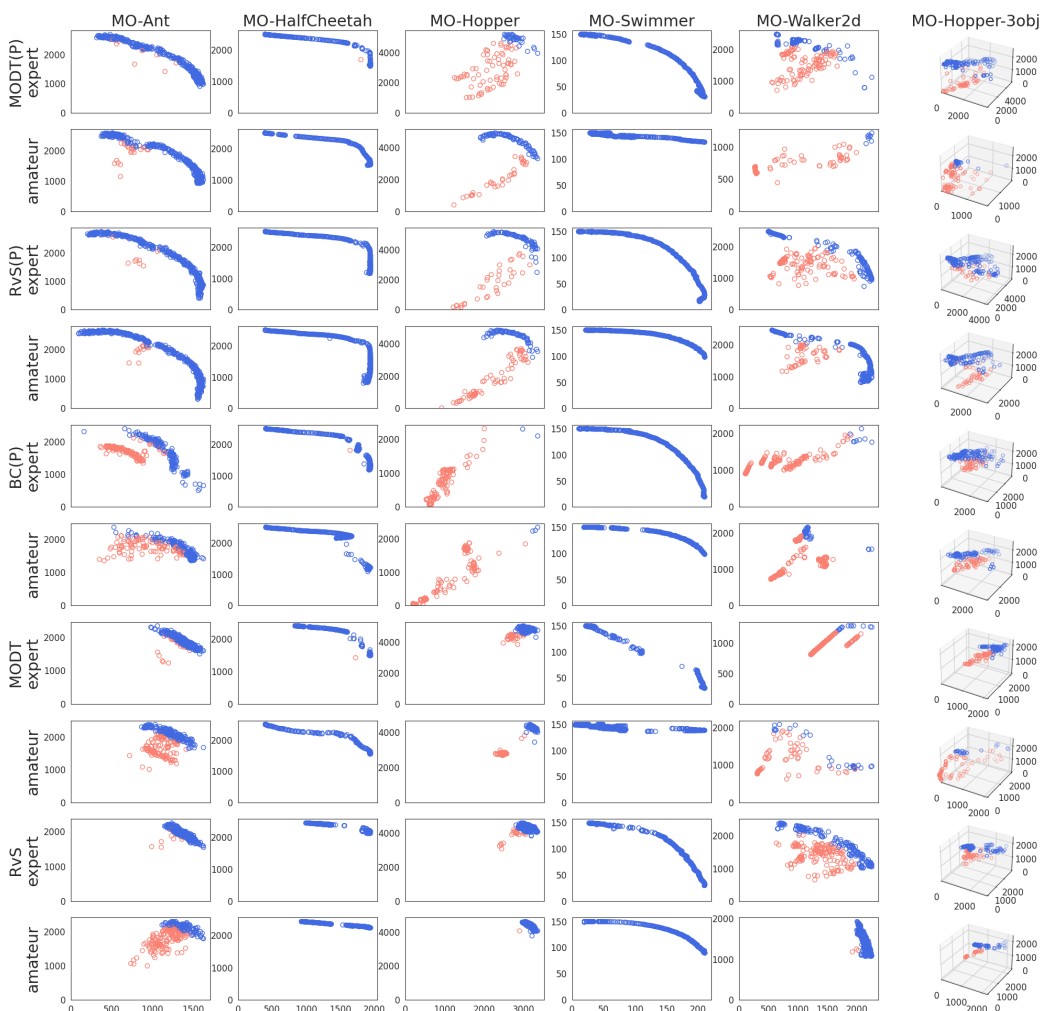

Figure 7: Pareto Visualization for all PEDA variants and baselines on `High-H` datasets. Notice that MO-Hopper and MO-Walker2d are more challenging and have significantly more dominated points for all variants. In other environments, however, the PEDA variants produce better results while other baselines fail at higher chances. Since BCwithout preference is completely single objective, we don't show the result here.

Table 6: We ablate how well PEDA variants perform and generalize under a different preference distribution in the MO-HalfCheetah environment. We can see that PEDA can still perform well when trained under the partially-clustered `Med-H` dataset. However, performance drops when it is trained under the entirely clustered `Low-H` dataset. (B: Behavioral Policy)

| Dataset ($\times 10^6$) | B | MODT(P) | MORvS(P) | BC(P) | MODT | MORvS | BC |
|---|---|---|---|---|---|---|---|
| `Med-H-Amateur` | 5.69 | 5.68±.01 | **5.77±.01** | 5.44±.26 | 5.61±.01 | 4.67±.03 | 3.21±.33 |
| `Low-H-Amateur` | 4.21 | **4.86±.05** | **4.80±.03** | 4.75±.04 | 4.72±.05 | 4.39±.04 | 4.05±.07 |
| `High-H-Amateur` | 5.68 | 5.69±.01 | **5.77±.00** | 5.46±.02 | 5.54±.02 | 4.66±.05 | 2.92±.38 |

Table 7: We ablate the importance of using multi-dimensional rtgs instead of a one-dimensional rtgs by taking the weighted sum of objectives on the MO-HalfCheetah environment. We see multi-objective rtgs provide a variance reduction for MORvS(P) and hypervolume performance improvement in other models. (B: Behavioral Policy)

| Setting | Dataset ($\times 10^6$) | B | MODT(P) | MORvS(P) | MODT | MORvS |
|---|---|---|---|---|---|---|
| mo rtg | `High-H-Expert` | 5.79 | 5.73±.00 | **5.78±.00** | 5.59±.03 | 4.19±.74 |
| 1-dim rtg | `High-H-Expert` | 5.79 | 5.70±.02 | **5.78±.00** | 4.43±.09 | 2.89±.06 |

datasets, which showcases the effectiveness of PEDA as an efficient MORL policy. Results are based on an average of 3 seeds with the standard error given.

# I TRAINING WITH 1-DIM RTG

We attempted to train MODT and RvS with 1-dim return-to-go rather than a separate rtg for each objective. According to results on MO-HalfCheetah and the `High-H` datasets in 7, using multi-dimensional RTG enhances the performance of MODT(P), and are about the same for MORvS(P) when preference is concatenated to states. However, it reduces standard error significantly in both MODT(P) and MORvS(P). In the naive models when preferences are not concatenated to states, using a multi-dimensional RTG helps to achieve a much more competitive hypervolume. We thus believe multi-dimensional RTG conveys important preference information when the model doesn't directly take preference as an input. Results are based on an average of 3 seeds with the standard error given.

