# OpenReview forum: "Scaling Pareto-Efficient Decision Making via Offline Multi-Objective RL"
_ICLR.cc/2023/Conference — ICLR 2023 poster_

### Official Review · Reviewer_gXRB · 2022-10-21

**Confidence:** 3
**Correctness:** 3
**Technical Novelty And Significance:** 3
**Empirical Novelty And Significance:** 4
**Recommendation:** 8

**Clarity, Quality, Novelty And Reproducibility:**

Clarity: Good
Quality: Good
Novelty: Reasonable
Reproducibility: Work should be reproducible.

**Strength And Weaknesses:**

Strength:
* The paper is well written and clear.
* The benchmarking dataset is a useful contribution for the community.
* Related work is well covered.
* The paper contains very useful illustrations, such as Fig 1 to illustrate Sec. 3, and Figures 2 and 3 to illustrate the data generation process for the benchmark.
* Extensive experiments. The generalization performance in Fig. 5 is very impressive.

Weaknesses:
* For data generation of the amateur policy, you set the exploration parameter p to 65%. This seems rather high to me: in most problems you will not get very far with such high exploration. Couldn’t you vary the amount of exploration, from low to high, for different episodes? Or gradually increase the noise during the episode, to make sure that your amateur policy sometimes gets a bit further in the domain? I think you see this effect in Fig 2: in MO-Swimmer for example you see that the amateur policies cover a quite demarcated area of the return space, indicating that your exploration/noise scheme covers a too small region of the overall return space.
* You extensively mention Decision Transformers in your abstract and introduction, but actually your best performing models are the MORvS(P) ones, that do not use a transformer. I think you would need to phrase this differently.
* On the algorithmic side, the innovation is mostly in the application of the (return-conditioned) sequence modeling approach to the offline RL setting. There are some details about how to feed the preferences and returns into the model, with some claims about what worked and did not work (without results though). It is a useful insight to use sequence modeling for offline RL though, since you typically want to stay close to the original data.
* Top of page 7: You say that a scalarized \hat{g} and the preference vector omega can recover the vector valued g. This is not true right? Imagine omega = [0.5,0.5], and \hat{g] = 0.5, then both g = [1,0] and g=[0,1] would work (or any g for which sum(g) = 1.0). I think this explains why you need the elementwise product between g and omega (but why not feed them in completely separately?).

Minor:
* Sec 2: previous work is “primarily” demonstrated on tabular tasks → But not only right? Try to be precise here, i.e., what is your extension?
* Sec 3: I miss an explanation why “sparsity” is a relevant measure?
* What type of noise distribution do you inject in generating the amateur policy? These are continuous tasks right, so do you use Gaussian noise, or simply a uniform distribution within the bounds of the action space?


**Summary Of The Paper:**

This paper studies offline multi-objective reinforcement learning (offline MORL). In the first half of the paper, the authors introduce a new benchmarking dataset. For 6 tasks it contains trajectories with different preferences, where for each preference we have data from an expert policy (with similar preference profile) and a noisy amateur policy. In the second half of the paper, the authors introduce a new approach for offline MORL, based on return-conditioned sequence modeling. Experiments show that their method outperforms baseline approaches on their own benchmark.

**Summary Of The Review:**

This paper has two contributions: 1) a new benchmarking dataset for offline multi-objective RL, and 2) a new algorithm class for offline MORL based on return-conditioned sequence modeling. I think the first aspect is a clear contribution, which will be of merit to the community. The second part is also interesting although slightly incremental. However, together I think the paper is a good candidate for acceptance at ICLR.

---

> ### Author Response · Authors · 2022-11-13
> **Thank you for your review!**
>
> Thank you for the careful reading of our paper and for providing helpful feedback!
>
> > **Couldn’t you vary the amount of the exploration from low to high, for different episodes? Or gradually increase the noise during the episode … MO-Swimmer for example you see that the amateur policies cover a quite demarcated area of the return space.**
>
> We agree there are many ways to generate suboptimal data. Our proposed methodology was based on 2 criteria: (a) Amateur returns should be sufficiently distinct from Pareto-optimal returns to explicit test for learning algorithms that can learn from suboptimal demonstrations; (b) Amateur returns should reflect non-zero training signal (i.e., not correspond to a random policy with very low returns) such that there are meaningful patterns for policy optimization. For the MO-Swimmer and other environments, we tried heuristic variants (e.g., using a mixture of random and expert policies at different ratios). However, overall our proposed strategy discussed in the paper worked best empirically in terms of satisfying the above desiderata.
>
> > **You extensively mention Decision Transformers in your abstract and introduction, but actually your best performing models are the MORvS(P) ones, that do not use a transformer.**
>
> We considered MODT(P) and MORvS(P) to be variants of the same family because they are both offline algorithms conditioned on future returns and chronologically, DT was the precursor to RvS. However, we agree that we should emphasize RvS more since MORvS(P) is the best-performing policy overall. We have made this modification in our updated version.
>
> > **Top of page 7: You say that a scalarized $\hat{g}$ and the preference vector omega can recover the vector valued g. This is not true right?**
>
> Thank you for pointing this out – it is a typo in our paper where the correct claim should be: “In principle, the scalarized RTG $\hat{g}$ can be recovered by the preference vector **$\omega$** and the vector-valued RTG **$g_t$**.” Empirically, we want to explain that conditioning on the vectorized return-to-go and preference provides only additional information for the model, as shown by the scalarized return-to-go used in single-objective DT and RvS can be recovered by our inputs. We have corrected this typo in our updated version.
>
> > **Why not feed rtg and preference completely separately without the elementwise product?**
>
> In principle, we agree that with more tuning and optimization tricks, feeding RTG and preference as tokens could also work. We have tried different model designs including the one proposed by the reviewer. We empirically find that the weighted  RTG worked best as a conditioning signal for the PEDA variants. In principle, we want to enforce two goals: (1) the ratio of RTG between each objective should match the respective ratio of preference; (2) when decrementing the RTG for each objective, the new RTG vector should preserve the same ratio between each objective. To achieve these criteria, the best practice we found is to use the weighted RTGs.
>
> > **Minor concerns on tabular tasks, sparsity metric, and noise distributions:**
>
> 1) Thanks for pointing it out. To the best of our knowledge, all of the previous works in offline MORL were only evaluated on environments with discrete actions.
> 2) The sparsity metric evaluates the density of the Pareto Front spanned by the undominated solutions. We compute sparsity by finding the average distance between the sorted undominated solutions.  Low sparsity indicates that the solutions are relatively well-spread over the frontier. We want to clarify that the sparsity metric is generally a secondary criterion for assessing models with high hypervolumes. This is because it is trivial to have good sparsity with a bad hypervolume, such as a policy that provides the same solution for distinct preferences.
> 3) We injected uniform noise to the actions to generate the Amateur dataset. Please refer to Appendix C for details. All environments we tested in this paper have continuous action spaces.
>
> Thank you again for your very helpful comments! Please feel free to follow up with any additional questions!

---

### Official Review · Reviewer_wtvp · 2022-10-23

**Confidence:** 4
**Correctness:** 3
**Technical Novelty And Significance:** 2
**Empirical Novelty And Significance:** 2
**Recommendation:** 5

**Clarity, Quality, Novelty And Reproducibility:**

The paper is well-written and mostly easy to follow.
Though multi-objective RL wasn't examined in the offline setting on the given environments before, the paper's novelty is limited. The technical contribution consists of feeding the preference as an extra input token to the method.
The paper provides a variety of information, and thus, I expect the reproducibility is quite high.

**Strength And Weaknesses:**

Strength:
* The paper is easy to follow.
* The paper examines the questions of how well decision transformers (and RLvSL) can be adapted to the multi-objective setting
* Multi-objective offline RL was not examined on Mucojo environments before.

Weaknesses:
* The proposed extension of the offline RL methods is relatively straightforward.
* Results are not too surprising.
* The novel data set is helpful, but given that the online benchmark was already available it is not too difficult to sample an offline data set.

**Summary Of The Paper:**

The paper proposes a reference benchmark for multi-objective offline learning data set which is built on a formerly proposed set of benchmark environments for multi-objective RL consisting of 6 MuCojo environments.
In addition, the authors adapt decision transformers and the RL via Supervised learning method to the multi-objective setting. This is achieved by concatenating the preference vector to state, action and return. Furthermore, the return to go is modified to represent the scalarized return.
In the experiments, the authors compare to behavioural cloning and conservative Q-learning which is modified to receive the input as preference as well. Results indicate that the supervised learning-based offline-RL approaches outperform compared methods.

**Summary Of The Review:**

The paper is a solid study of offline multi-objective RL on the MuCojo settings. However, the paper lacks novelty and technical depth. All solutions are kind of relatively straightforward and as compared methods do not fit the problem results are not too convincing. The sampling scheme to generate the offline data seems solid but sampling the data did not require overcoming any technical problems.
To conclude, the paper yields interesting results but does not come up with the technical depth and novelty of most accepted ICLR papers.

---

> ### Author Response · Authors · 2022-11-13
> **Thank you for your review!**
>
> We would first like to appreciate your time in reading our manuscript and providing helpful feedback!
>
> The main concern of the reviewer is “Limited novelty/technical depth” as reflected in all the weaknesses that consider our approach and result straightforward. To this end, we would like to highlight that this is one of the first works in multi-objective offline RL. While the results might look unsurprising in hindsight, we believe they are partly a reflection of our extensive and rigorous study of dataset and model designs.  We expand on these points below:
>
> **Dataset Design:** Datasets are critical for the standardized evaluation of offline RL. Even when given a set of reference policies, we believe it is not obvious how the datasets should be constructed. As discussed in the paper, we have formalized design choices relating to the optimality of the behavioral policies (amateur/expert) and prior distributions on preferences (based on entropy values). These design choices provide key insights into the generalization performance of downstream algorithms to in-distribution and out-of-distribution preferences. Moreover, we considered the Gym suite of environments due to their widespread use in the RL community, thus reducing the bar for entry and promoting future research using the proposed datasets.
>
> **Model Design:** We extended previous offline single-objective algorithms to design PEDA variants. In doing so, we studied design choices that enable successful extension such as the use of preference tokens (e.g., state concatenation vs. separate tokenization), as well as the use of multi-objective return targets. We also compared different base algorithms (RvS, DT, CQL) in the multi-objective offline setting and our relative trends are also somewhat different from those reported in the single-objective setting for many environments (e.g., MO-RvS outperforms MO-CQL). In terms of absolute performance, we showed that PEDA variants can generalize well even when datasets are suboptimal (see e.g., Table 2).
>
> Thank you again for your feedback and please feel free to follow up with any additional questions or concerns!

---

### Official Review · Reviewer_6PPS · 2022-10-25

**Confidence:** 4
**Correctness:** 3
**Technical Novelty And Significance:** 2
**Empirical Novelty And Significance:** 2
**Recommendation:** 6

**Clarity, Quality, Novelty And Reproducibility:**

Overall the paper seems fairly well written and organized. The approach seems fairly novel (though incrementally so). Reproducibility seems like it might be challenging from the paper alone, but not egregiously outside the current standard for ML publications.

**Strength And Weaknesses:**

strengths
-------------
- An interesting approach to an interesting (and nontrivial) problem.

- The idea of a sample-efficient way for approximating the Pareto front at execution time seems quite promising.

Weaknesses
---------------
- The dataset itself doesn't seem like that much of a contribution, compared to how it is framed in the paper.

- Adding random perturbations to expert demonstrations is not a good way of approximating amateur demonstrations.

- It's unclear how this translates to a more organic multi-objective setting (consider a navigation task where vehicles need to maximize transport throughput while minimizing risk and travel time).

- Not clear how well this would generalize to remote parts of the trajectory space (and preference space).

**Summary Of The Paper:**

In this paper the authors propose a flexible multi-objective RL framework that can handle different preferences over objectives, that are not known in training time. They achieve this with an offline RL approach. First, they present a novel dataset of demonstrations designed for offline multi-objective settings. Second, they propose a Pareto-efficient decision mechanism that builds on Decision Transformers with a novel approach for conditioning the policies on preferences and returns. The authors continue to show that their proposed framework closely matches the behavior policies in the demonstrations, meaning it effectively approximates the Pareto front.

**Summary Of The Review:**

Overall an interesting paper with some caveats that need addressing.

---

> ### Author Response · Authors · 2022-11-13
> **Thank you for your review!**
>
> We appreciate your time and helpful comments on our paper!
>
> > **The dataset itself doesn’t seem like much of a contribution:**
>
> Our datasets are indeed one of our key contributions, as the community currently lacks such a benchmark even when the problem setting is quite important from a practical perspective. Moreover, we spent considerable thought in ensuring that the utility of the datasets extends beyond the current work to other researchers. First, we are focusing on the widely-used and standardized Gym environments used in the majority of papers in deep reinforcement learning. Secondly, the D4RL offline dataset and benchmark proposed by [Fu et al. 2020] for the single-objective setting have enjoyed widespread use by the Offline RL sub-community. Finally, we want to emphasize that D4MORL is the first offline dataset for MORL, and we believe our range of design choices (e.g., expert/amateur, different preference priors) in generating different dataset variants will collectively aid rigorous strength-testing of any future algorithms.
>
> > **Adding random perturbations to expert demonstrations is not a good way of approximating amateur demonstrations:**
>
> We agree that there are potentially different ways to generate amateur demonstrations and this is an important direction for future study. Our proposed methodology was based on satisfying 2 desiderata: (a) trajectory returns should be sufficiently distinct from Pareto-optimal returns to explicit test for learning algorithms that can learn from suboptimal demonstrations; (b) trajectory returns should reflect non-zero training signal (i.e., not correspond to a random policy with very low returns) such that there are meaningful patterns for policy optimization. It is hard to reverse engineer policies that will satisfy the above constraints and we tried some heuristic variants (eg, using a mixture of random and expert policies); however, overall our proposed strategy discussed in the paper worked best empirically in terms of satisfying the above desiderata.
>
> > **Not clear how well this (PEDA) would generalize to remote parts of the trajectory space (and preference space):**
>
> We believe that experiment results in Med-H and Low-H preference distributions from Table 6 in Appendix G would provide helpful information for generalizability to preference space. As for generalizability to remote parts of the trajectory space, we believe results in Figure 5, which show that the agents closely follow the conditioned reward for each objective, even if the initial target is out of the dataset minimum and maximum range, would be helpful.
>
> Thank you again for your time and helpful comments! Please feel free to follow up with any additional questions.

---

### Official Review · Reviewer_oRVV · 2022-10-30

**Confidence:** 4
**Correctness:** 4
**Technical Novelty And Significance:** 3
**Empirical Novelty And Significance:** Not applicable
**Recommendation:** 6

**Clarity, Quality, Novelty And Reproducibility:**

The presentation is in high clarity and quality. See the previous part for novelty. I have no tell on its reproducibility.

**Strength And Weaknesses:**

Pro:

1. This paper introduces a new dataset that targets the multi-objective RL with offline data.

2. This paper introduces a decision transformer-like algorithm which is intuitive and works in experiments.

Cons:

The baselines in the experiments seem lacking as only two methods (BC/CQL) are compared with. Is any offline RL and inverse RL approach relevant?

Questions

1. One question is how the two contributions - namely the dataset and the algorithm - are entangled to each other. With the presentation of this paper one would expect the contributions to be independent, i.e. we expect the algorithm to work on other datasets as well. Is there any way to validate this? I understand a similar dataset might not be present in the community for now.

2. Following the last question, is the dataset general enough to be an independent contribution? From the generation of this dataset I could many variants in the process. Should the community use this dataset in the future or it's good only for this work.

3. What is the difference of this work and multi-objective inverse RL? I'm not seeing much difference despite the work follows the vein of offline RL mostly.


**Summary Of The Paper:**

This manuscript did two things. One is to propose a new dataset that collects agent trajectories from multiple independent agents with different preferences on the objectives. This dataset contains trajectories from both well-trained agents and semi-trained agents. HalfCheetah is a typical example where actions that are large in absolute value will consume more energy while the agent wants the cheetah to run as fast as possible. Two is to propose a decision transformer-like algorithm that handles an offline multi-objective dataset (which is possibly the one proposed). This algorithm is quite intuitive and the main idea is to condition its output on the preference of the newly tested task. Several experiments are presented.



**Summary Of The Review:**

I think it's quite an interesting problem and the work is solid (intuitive, works well, and is presented well). I believe it's a borderline paper and I lean slightly towards acceptance.

---

> ### Author Response · Authors · 2022-11-13
> **Thank you for your review!**
>
> Thank you for your helpful comments on our paper!
>
> >**How the dataset and the algorithm are entangled to each other?**
>
> Indeed, the two contributions are of independent interest. As the reviewer acknowledges, while the problem is interesting and practically relevant, we were concerned to find that there are no standard offline datasets for evaluating and benchmarking offline MORL algorithms in non-tabular MDPs. Our class of algorithms in PEDA are however general purpose as they do not make assumptions that are specific to any of the environments in D4MORL, and we are confident that PEDA should work on other datasets as well!
>
> >**Is the dataset general enough to be an independent contribution? …Should the community use this dataset in the future?**
>
> Yes, we believe the datasets in D4MORL are general enough as an independent contribution and can be used by the community for future study. The underlying environments are ones derived from the Gym suite implemented using the MuJoCo simulator – this suite is well-acknowledged in the deep reinforcement learning community and has been used in several papers. Moreover, even the D4RL benchmark [Fu et al., 2020] that proposed offline datasets for the single objective setting is primarily designed for these Gym environments and has enjoyed widespread use in the offline RL literature. Finally, we considered a wide range of design choices (e.g., expert/amateur, different preference priors) in generating different dataset variants that we anticipate will collectively aid rigorous strength-testing of any future algorithms.
>
> >**What is the difference of this work and multi-objective inverse RL?**
>
> The goal of multi-objective inverse RL is to *infer the rewards and/or preferences* of the different objectives being optimized by an agent. In contrast, we are in the offline multi-objective RL setting where the rewards and preferences are given in the training set of demonstrations. The goal in our case is to *learn a preference-agnostic policy* that can be used at test-time to optimize an arbitrary set of user preferences. While different from our current setting, we agree that multi-objective inverse RL is an exciting direction for future work as current works in MO-IRL (Kishikawa et al., Takayama1 et al.) have yet not scaled to continuous state and action spaces, unlike our current work.
>
> Thank you again for your comments, and please feel free to follow up with any other questions and concerns!
>
> ---
>
> >References:
>
> >Takayama, N., Arai, S. Multi-objective deep inverse reinforcement learning for weight estimation of objectives. Artif Life Robotics 27, 594–602 (2022). https://doi.org/10.1007/s10015-022-00773-8
>
> >D. Kishikawa and S. Arai, "Multi-Objective Inverse Reinforcement Learning via Non-Negative Matrix Factorization," 2021 10th International Congress on Advanced Applied Informatics (IIAI-AAI), 2021, pp. 452-457, doi: 10.1109/IIAI-AAI53430.2021.00078.

---

> > ### Comment · Reviewer_oRVV · 2022-12-01
> > **Thank you for your response**
> >
> > I thank the authors for their response. As the response and my review seem to be on the same page, I will maintain my current review and rating.

---

### Decision · Program_Chairs · 2023-01-20

**Decision:**

Accept: poster

**Justification For Why Not Higher Score:**

Landing an offline RL datasets for multi objective RL can potentially have great impact to the community. Unfortunately reviewers are not fully convinced if the dataset is representative of the multi objective setup in real world. The algorithm is straightforward, which I consider a plus. However, the enceignants miss any comparison to multi objective RL baselines.

**Justification For Why Not Lower Score:**

The paper is clearly written. Reviewers are landing on the positive side.

**Metareview: Summary, Strengths And Weaknesses:**

The paper made two contributions: 1) a multi objective offline RL dataset, D4MORL; 2) a decision transformer based algorithm to the multi-objective setting,  by concatenating the preference vector to state, action and return. The algorithm is straightforward and shows promise on the proposed datasets. The paper is clearly written. Reviewers are landing on the positive side without strong champions though. Landing an offline RL datasets for multi objective RL can potentially have great impact to the community. Unfortunately reviewers are not fully convinced if the dataset is representative of the multi objective setup in real world. The algorithm is straightforward, which I consider a plus. However, the experiments miss any comparison to multi objective RL baselines.

**Note From Pc:**

if the above contains the word "oral" or "spotlight" please see: "oral" presentation means -> notable-top-5% and "spotlight" means -> notable-top-25%. As stated in our emails, we are disassociating presentation type from AC recommendations